# Chest drain REgular FLushing in CompIIcated parapneumonic EFfusions and empyemas: Study protocol for the RELIEF randomized controlled trial

Taryn K. Boyle[1◉], Jennifer D. Duke[1◉], Gulmira Yermakhanova[1], Rafael Paez[1], Greta Bridwell[1], Ankush P. Ratwani[2], Kaele M. Leonard[1], Heidi Chen[3], Frank E. Harrell Jr[3], Robert J. Lentz[1], Fabien Maldonado[1], Najib M. Rahman[4], Samira Shojaee[1*]

1 Division of Allergy, Pulmonary and Critical Care Medicine, Vanderbilt University Medical Center, Nashville, Tennessee, United States of America, 2 Creighton University, Division of Pulmonary, Critical Care, and Sleep Medicine, Omaha, Nebraska, United States of America, 3 Department of Biostatistics, Vanderbilt University Medical Center, Nashville, Tennessee, United States of America, 4 Oxford Respiratory Trials Unit, Oxford NIHR Biomedical Research Centre for Respiratory Medicine, Nuffield Department of Medicine, University of Oxford, Oxford, United Kingdom

◉ These authors contributed equally to this work.
* samira.shojaee@vumc.org

## Abstract

### Background

Pleural infections are common and drainage of the pleural space, in addition to antimicrobial therapy, is often required for adequate treatment. Guidelines suggest flushing small bore chest drains with 20–30 mL of saline every six hours, however, no randomized controlled trials (RCTs) have assessed if this practice improves outcomes for pleural space infections. As a result, flushing practice is varied, inconsistent, and confounds the interpretation of studied therapeutic modalities in pleural space infection trials. The impact of regular chest drain flushing compared to as-needed flushing on length of time to chest tube removal is unclear.

### Methods

Chest Drain **RE**gular F**L**ushing in CompI**I**cated Parapneumonic **EF**fusions and Empyemas (RELIEF) is a multi-center, open label randomized controlled trial conducted in the United States. Patients with a pleural space infection requiring chest drain placement for inpatient management will be screened for eligibility. Patients will be randomized within 24 hours of chest drain placement to a regular flushing protocol versus as-needed flushing for drain blockage. The primary outcome is time from randomization until time to chest drain removal (hours). Secondary outcomes are length of hospitalization, degree of radiographic improvement by chest X-ray, ultrasound or CT scan from time of drain placement to time of removal, need for additional

purpose. The work is made available under the Creative Commons CC0 public domain dedication.

**Data availability statement:** No datasets were generated or analysed during the current study. All relevant data from this study will be made available upon study completion.

**Funding:** This study is funded by Cook Medical, Inc. The funders had no role in study design, data collection and analysis, decision to publish, or preparation of the manuscript.

**Competing interests:** Samira Shojaee: Research funding from Cook Medical. Fabien Maldonado: consulting for Medtronic, Intuitive, and J&J, Principal Investigator for the ACES study with research funding from Pleura Dynamics. Research funding from Medtronic. Robert Lentz: consulting for Intuitive. Jennifer Duke: consulting for Intuitive, research funding from Cook Medical.

procedures for the management of pleural space infection, and complications. An ordinal, multi-state transition model will be used to precisely characterize the role of flushing in longitudinal clinical outcomes in the two arms.

## Discussion

RELIEF is a multi-center, open label randomized controlled trial that compares a regular saline flushing protocol with as-needed saline flushing of small-bore chest drains (8–20Fr) for the management of pleural space infection. This will be the first randomized controlled trial evaluating flushing protocol with patient-centered outcomes in pleural space infections.

## Trial Registration

The trial was registered in ClinicalTrials.gov (NCT06427538) on 05-10-2024.

## Background

Pleural infection occurs in an estimated 65,000 patients per year in the U.S. and UK [1] and is associated with substantial morbidity and mortality [2]. The most recent BTS guidelines recommend that patients with frankly purulent, turbid/cloudy pleural fluid, or suspected complicated parapneumonic effusion with suspicious pleural fluid laboratory and radiographic features receive prompt pleural space drainage with a small-bore chest drain (<14Fr) [3].

One downside of smaller bore chest drains for pleural infection management is the perception of a heightened risk of drain obstruction, which may be ameliorated via flushing of the catheter with sterile saline and suction application. The 2010 BTS guidelines recommended regular flushing (20−30 mL saline every 6 hours via a three-way stopcock for small-bore catheters and the application of suction (−20 cm $H_2O$) in the hopes of improved drainage [4]. Most studies of small-bore catheters have used both flushing and suction to decrease the likelihood of catheter blockage and improve drainage efficiency. However, the role of this practice in successful drainage of infected fluid has never been studied prospectively [5,6]. The most recent 2023 BTS guideline update does not address optimal saline flush management of chest drains [3].

Previous studies demonstrate inconsistent protocols and findings regarding the optimal utilization of saline flushing in chest drain management. One retrospective review of one-hundred 12 Fr chest drains (20% placed for pleural infections) found that 9% of the drains became blocked [6]. Fifty-eight of these catheters were flushed regularly with 20 mL of sterile saline every 6 hours, resulting in lower rate of drain blockage (odds ratio (OR) for blockage in flushed drains vs non-flushed drains 0.04, 95% CI: 0.01–0.37, p < 0.001) [6]. Conversely, Horsley et al. prospectively analyzed 52 catheters (ranging from size 12–20 Fr) and found that 6 of 10 drains placed for empyema management were blocked. Regular flushing with 30 mL saline was used but did not appear to alter the rate of blockage [7].

Many centers within and outside the US do not follow regular flushing protocols and instead, evaluate for blockage and flush if a chest drain appears blocked and non-draining, given minimal supporting data for either practice. In the observational retrospective study by Cafarotti et al., ninety-seven 12Fr drains were placed for empyema, with 85% of the 72 drains removed due to drainage failure cited as secondary to blockage, often with observed fibrin or blood clot in the drain. Drain flushing was performed with 50 mL of saline solution when blockage was suspected [5]. Variation in chest drain flushing practice is often cited as a study limitation in literature surrounding pleural space infection management. Given the absence of rigorous studies guiding optimal drain management, we propose RELIEF, a multicenter trial evaluating the utility of regular flushing of chest drains placed for the management of pleural space infection.

## Methods

### Trial design

This study is a prospective, multicenter, open label, randomized controlled trial. The primary objective of this study is to evaluate the impact of regular chest drain flushing on the time to drain removal for patients hospitalized with an infected pleural space requiring chest drain placement.

Secondary objectives of this study include evaluation of the impact of regular chest drain flushing on successful or failed outcomes over the chest drain dwell period, as well as the total length of hospitalization stay, degree of radiologic improvement, need for additional pleural interventions, and associated complications of chest drain management. Regular saline flushing (e.g., 20–30 mL saline every 6 hours via a three-way stopcock) is recommended for small-bore chest drains placed for pleural infection drainage [3]. Most studies of small-bore chest drains have used both flushing and suction to decrease the likelihood of drain blockage and improve drainage efficiency, without a standardized protocol. Our comparators reflect the practice variation in chest drain flushing across centers by comparing a regular flushing protocol to as needed flushing in chest drains placed for draining of pleural infection.

This study has been approved by the Institutional Review Boards (IRB) at Vanderbilt University Medical Center (IRB# 231367), Virginia Commonwealth University (IRB #HM20031525), Mount Sinai Hospital (IRB# 24–01303), Creighton University (IRB # 2005001) and Henry Ford Health (IRB #18616). The trial is registered on ClinicalTrials.gov (NCT06427538) prior to the study opening.

### Study setting

The primary study site is Vanderbilt University Medical Center, an academic tertiary care hospital in Nashville, TN (start date: 6/21/2024). Current secondary study sites include other academic tertiary care centers within the United States: Virginia Commonwealth University (start date: 3/18/2025), Creighton University (start date: 6/24/2025) and Henry Ford Health (start date: 7/29/2025). Participant recruitment is expected to be completed by 12/2026. Data collection is expected to be completed by 2/2027 and results are expected in 7/2027.

### Study population

All adult inpatients, 18 years or older, requiring chest tube placement for the management of pleural infections at participating sites are eligible.

### Informed consent

Research informed consent will be obtained by a study team member within 24 hours of chest drain placement.

## Randomization and blinding

Patient level 1:1 randomization will be generated via REDcap (Research Electronic Data Capture) electronic data capture tools hosted at Vanderbilt University (www.project-redcap.org) [8,9]. Allocation is concealed in this online, restricted-access database.

Intervention begins at time of randomization, which is protocolized to be within 24 hours of chest drain placement for pleural infection. The intervention discontinues at the time of chest drain removal for successful treatment of pleural space infection, or uncommon instances when the chest drain ceases to be utilized for the drainage of a pleural space infection but should remain in situ (for example, a resolved pleural infection complicated with a bronchopleural fistula with pneumothorax requiring chest tube drainage of air). Chest drain removal due to dislodgement, malposition, or "kinking", or if the patient is deceased with a chest drain in situ are considered competing risks. If additional pleural infection treatments are deemed necessary (including additional chest drain placement, or surgical treatment), the chest drain of interest is considered a treatment failure, resulting in discontinuation of the allocated intervention.

Because the study investigators and participants are not blinded to the study arm, specific objective criteria for chest drain removal due to successful pleural space infection management will be assessed on a daily basis. These criteria include: 1) absence of (or minimal free flowing) pleural fluid on ultrasound or chest X-ray, AND 2) no concern for an ongoing pleural infection that requires continuous drainage, AND 3) overall clinical judgement is that the chest drain is ready to be removed. The chest drain removal data are documented on a daily basis, to identify changes in the behavior of providers due to lack of blinding, and for sensitivity analysis. If the chest drain is required to remain in situ for other reasons, the investigator documents day and time of chest drain removal criteria met, and reasons for the drain remaining in situ. Chest X-ray imaging data will be interpreted by radiologists blinded to the patient arm, at the corresponding institution. Following trial completion, an adjudication committee blinded to intervention arm will review the clinical decision for drain removal due to the successful treatment of pleural infection (primary endpoint), as well as the categorization of drain status as "successful," "ongoing" or "failed" (secondary endpoint). Data will be obtained from Case Report Forms and imaging documentation on day of chest drain removal, and committee findings will be utilized for sensitivity analysis.

## Trial interventions

Patients in the intervention group will have 20 mL sterile saline flushed into their chest drains by trained nurses or doctors every 6 ± 2 hours. If patients are receiving tPA/DNase (administered at the same time, or in rapid sequence), each treatment will be considered one flush.

The control group will have no saline routinely instilled into the chest drain. If during daily assessment, the chest drain is obstructed, defined by lack of tidaling, lack of fluid drainage, and presence of fluid on US exam, it will be flushed with 20 mL of saline (Fig 1).

All chest drains in both arms will be assessed for blockage once per day, by study personnel (separate from the nursing or doctors who will instill saline in the intervention arm).

All patients in both arms will have −20 cm $H_2O$ at the atrium continuous suction (defined as ≥75% of the day with suction) applied to their drainage system at all times, if tolerated. Length of time with discontinued suction in the setting of chest discomfort or pain will be documented.

Research staff will directly monitor the daily adherence to the intervention during daily data collection. All patients enrolled and randomized to a study arm will be included in the intention-to-treat analysis.

We do not exclude patients in either arm that are unable to tolerate suction after initial application (for example, a patient with trapped lung and excess pain during the application of suction) or have suction disconnected for a period of time (for example, for patient mobility or transport, etc.) and feel this reflects most accurately the logistics of clinical care

 

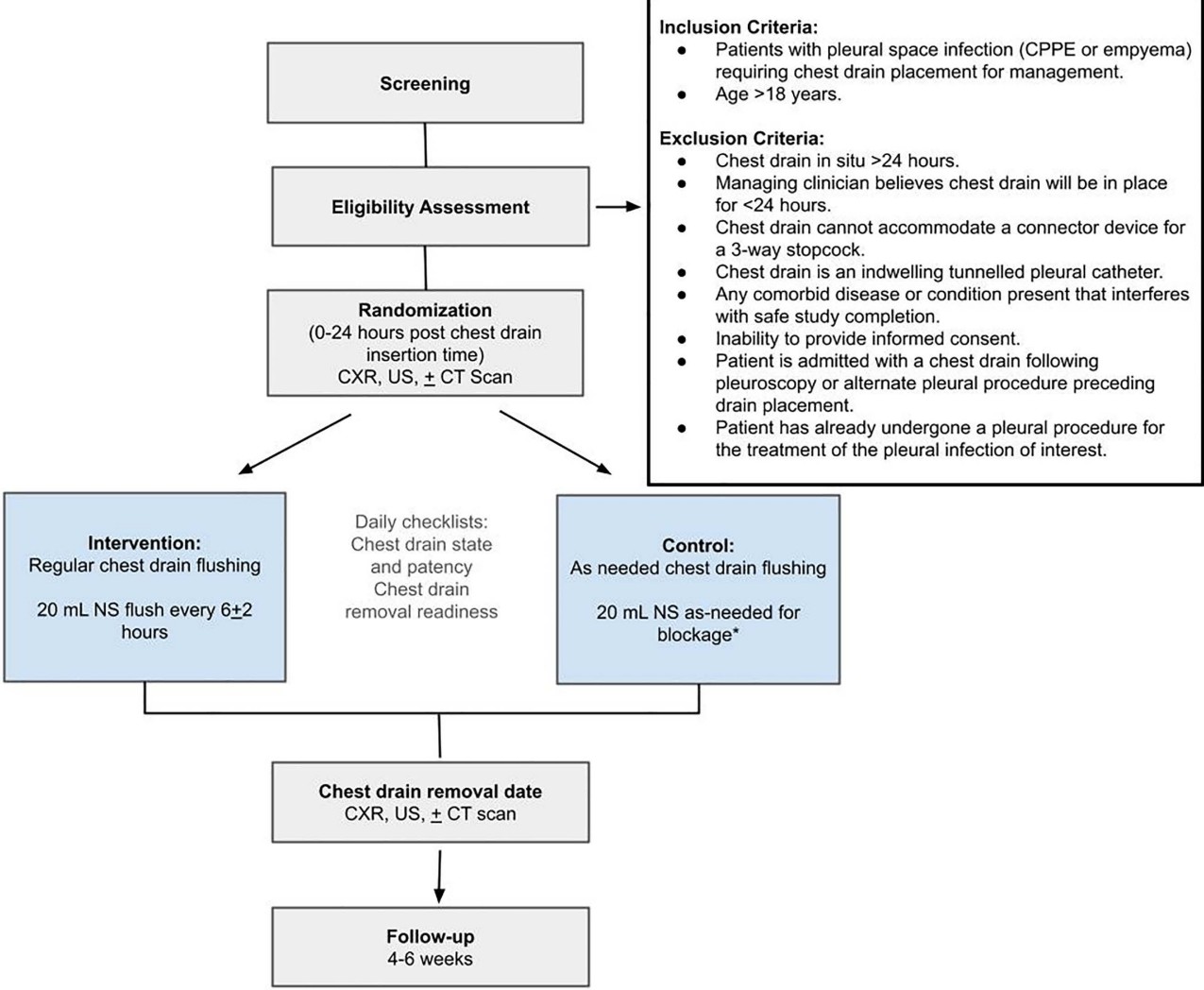

**Fig 1. Schematic of study design.** Abbreviations: CPPE: Complicated parapneumonic effusion, IET: Intrapleural enzyme therapy (tPA/DNase), NS: Normal saline, CXR: Chest X-ray, US: Ultrasound, CT scan: Computed Tomography scan. *Chest drain blockage is assessed by lack of tidaling, lack of fluid drainage, and presence of pleural fluid on US exam.

with chest drains. Intrapleural enzyme therapy may be administered if required based on the judgment of the primary provider [10,11], and one dosage of intrapleural enzyme therapy is considered as one flush.

This study does not influence the need for additional treatment options (additional chest drains or surgical management) assessed by the primary medical team. If additional procedures are required to manage the pleural infection, the index chest drain treatment will be considered as failed and ceases further intervention.

No additional ancillary care or follow-up beyond routine clinical care is planned for this trial.

## Outcomes

**Primary outcome.** The primary outcome is the time (hours) from randomization to chest drain removal, because of intentional clinical decision to remove the drain for treatment success.

**Secondary, safety and exploratory outcomes.** Secondary outcomes include:

1 – length of hospitalization (hours).

2 – radiographic improvement as evidenced by chest X-ray, ultrasound or CT scan at the time of chest drain placement compared to the time of removal.

3 – additional procedures for the management of pleural space infection (VATS, additional drains, etc.).

4 – Complications.

## Multistate transition models

A multistate model is a modeling framework for continuous processes that allows individuals to move among a finite number of "states," reflecting changes in disease status over time. Multistate modeling incorporates not only the first disease state, but also subsequent and multiple intermediate states that may or may not be reached until a patient reaches a final, "absorbing state."

Our data involves three processes: 1) whether patients received tPA/DNase therapy or not, 2) whether patient's chest drain was deemed clogged vs patent, <u>after</u> appropriate steps taken to unclog a drain and 3) whether the chest drain±tPA/DNase was considered a treatment failure, requiring additional invasive procedures. Patients could experience a clogged drain but continue to have ongoing treatment in the form of antibiotics, relief with rescue flushing and the use of tPA/DNase therapy, but if a patient was considered a treatment failure, this state will be defined as an absorbing state (a state in which the probability of leaving that state is zero). Another absorbing state is when a patient is considered a treatment success, where inpatient pleural infection management is complete and a chest drain for this purpose is no longer needed.

Considering these three processes yielded twelve possible states, or categories (see below): 1) patent chest drain with treatment successful, no tPA/DNase (in the past 24 hours), 2) clogged chest drain with treatment successful, no tPA/DNase, 3) patent chest drain with tPA/DNase, treatment successful, 4) clogged chest drain with treatment successful, with tPA/DNase, 5) patent chest drain with ongoing treatment, no tPA/DNase 6) patent chest drain with tPA/DNase with ongoing treatment, 7) clogged chest drain with ongoing treatment, no tPA/DNase 8) clogged chest drain with tPA/DNase with ongoing treatment, 9) patent chest drain with failed treatment, no tPA/DNase 10) patent chest drain with tPA/DNase with failed treatment, 11) clogged chest drain with failed treatment, with no tPA/DNase and 12) clogged chest drain with tPA/DNase with failed treatment.

We define four ordinal levels of status based on each category (Groups 1–4, Table 1) and each patient's status (success, ongoing, failed) will be assessed daily. At a given point in time (time of assessment), a patient can only be in one category; while reaching an absorbing state, defined as treatment success, or treatment failure (groups 1 or groups 4) results in a state with no further transitions possible.

## Definition of states

*Absorbing state* is a state in which the probability of leaving that state is zero and denotes treatment success or failure.
  *Treatment status:*

  - *Successful Treatment* notes inpatient effusion management is complete, as assessed by

  presence or absence of pleural fluid on US or CXR, concern for ongoing pleural infection and

  clinical judgement of readiness for removal by the treatment team. Successful treatment

  suggests a chest drain can be removed and the patient could be discharged from a pleural

**Table 1. Patient categories based on chest tube and treatment status organized into ordinal levels in a multistate modeling framework.**

| Categories | Drain Patency: Patent vs. Clogged | tPA/DNase in the last 24 hours | Outcome/Status: Ongoing, Success, Failure | Groups |
|---|---|---|---|---|
| 1 | Patent | No | Successful | 1 (absorbing) |
| 2 | Clogged | No | Successful | 1 (absorbing) |
| 3 | Patent | Yes | Successful | 1 (absorbing) |
| 4 | Clogged | Yes | Successful | 1 (absorbing) |
| 5 | Patent | No | Ongoing | 2 |
| 6 | Patent | Yes | Ongoing | 2 |
| 7 | Clogged | No | Ongoing | 3 |
| 8 | Clogged | Yes | Ongoing | 3 |
| 9 | Patent | No | Failed | 4 (absorbing) |
| 10 | Patent | Yes | Failed | 4 (absorbing) |
| 11 | Clogged | No | Failed | 4 (absorbing) |
| 12 | Clogged | Yes | Failed | 4 (absorbing) |

infection standpoint.

*- Ongoing Treatment* notes inpatient effusion drainage with the index chest drain is actively

underway.

*- Failed Treatment* notes the patient needs a procedure in addition to the index chest drain for management of their pleural space (e.g., additional chest drain, surgery, etc.).

*Drain Patency* is assessed by the presence of tidaling and evidence of ongoing drainage output (*clogged vs. patent)*.

## Recruitment

Every patient undergoing chest drain placement at Vanderbilt University Medical Center and participating sites will be screened for eligibility through an electronic medical record-based algorithm that alerts the team to chest drain placements. Once the investigator has determined the subject's eligibility for the study, patients are approached, and verbal and written informed consent is obtained. Subjects who fail to meet the entry criteria will be excluded from the study and considered a screen failure. Screen failures will be recorded without patient information. Subjects will be able to withdraw from the study at any point.

## Inclusion criteria

- Patients with pleural space infection (CPPE or empyema) requiring chest drain placement as standard of care for inpatient management of their pleural space infection.

- Age > 18 years old.

## Exclusion criteria

- Chest drain has been placed >24 hours.

- Patients who have chest drains that cannot accommodate a connector device for a 3-way stopcock.

- Chest drain is an indwelling tunneled pleural catheter.

- Any comorbid disease or condition that interferes with the safe completion of the study.

- Inability to provide informed consent.

- Managing clinician believes the chest drain will be in place for <24 hours.

- Patients admitted with a chest drain following thoracoscopy or other pleural procedure preceding drain placement.

- Patient has already undergone a pleural procedure for the treatment of the pleural infection of interest (ex: prior chest drain placement or surgical intervention).

### Study calendar

Patients will be eligible for randomization up to 24 hours after chest drain placement. Chest X-ray and ultrasound images obtained during usual clinical care at the time of chest drain placement will be reviewed and their information documented as baseline/day 1. The study team will obtain a baseline assessment of the pleural effusion via ultrasound if no ultrasound images are available within 24 hours prior to chest drain placement.

Subjects will be randomly allocated into intervention (regular flushing) and control (flush as-needed) groups using computer-generated randomization just prior to starting the procedure. A daily checklist will be completed by the research team to assess chest drain blockages and ongoing need for the tube/potential removal of the catheter (Fig 2).

On the day of chest tube removal, both the intervention group and control group will undergo a repeat chest X-ray. Post-chest drain removal ultrasound images will be obtained by the research team. Chest X-rays can be performed as portable semi-erect or PA and lateral based on the clinical judgment of the primary provider. A post hospital discharge follow-up visit will take place as standard of care management (4–6 weeks post-discharge, or earlier if required) as decided by the primary provider.

### Sample size calculation

Previous studies have shown a varying number of days to chest drain removal based on etiology, with a mean of 7.6 days (SD: 1.53) in a study of 98 patients with infected pleural space undergoing regular saline flushing [5]. There is limited data with only one small study reporting median days to chest drain removal (5 days (3–7) in a study of 36 patients with no-flush strategy) [7]. Assuming a non-normal distribution, with data skewed to the right, we estimated a median of 6 days to chest tube removal. Assuming a difference of 24 hours day (1 day), earlier chest drain removal in the intervention (144 hours/6 days in the SOC group and 120 hours, 5 days in the intervention group) is a clinically meaningful difference to detect. The sample size is calculated from a two-sided Wilcoxon rank-sum test at a 0.05 significance level with a standard deviation of 36 hours (1.5 days). Group sample sizes of 38 in each arm achieve 80% power to detect a median difference of 24 hours (1 day). The sample size estimation is based on a 2000 Monte Carlo simulation assuming samples from a range of distributions such as Gamma, Lognormal, and Gumbel distributions. Assuming a 10% withdrawal rate and 15% intrapleural enzyme therapy failure (which can prevent reaching the primary endpoint), we plan to enroll 96 participants (with randomization performed in a 1:1 ratio). To avoid an imbalance in randomization among groups, randomization will be stratified by intrapleural enzyme therapy use.

### Data collection

All data collected in this study are captured as part of routine clinical care. Data will be collected in a Health Insurance Portability and Accountability Act (HIPAA) compliant REDCap database [8,9]. The database is composed of several case report forms (CRFs) that include demographic, radiographic, procedure, complication, and follow up data. Data will be entered by the study team members who are trained in use of the CRF. Data collected in the CRFs include but are not limited to:

| Events | Within 24 hours post chest drain placement | Day 1 | Day 2 | Day 3 | Day 4 | Days until chest drain removal | Day of chest drain removal | 4-6 weeks follow up |
|---|---|---|---|---|---|---|---|---|
| Screening/Enrollment | X | | | | | | | |
| Informed Consent | X | | | | | | | |
| Randomization | X | | | | | | | |
| Chest X-ray* | X | | | | | | X | |
| Ultrasound exam | X | | | | | | X | |
| Intervention group (regular flushing) | | | | | | | | |
| Chest drain daily flushing (20 mL saline) | X** | X | X | X | X | X | | |
| Daily checklist for chest drain blockage assessment (states, patency, blockage) | | X | X | X | X | X | | |
| Daily checklist for chest drain removal assessment | | X | X | X | X | X | | |
| Follow up | | | | | | | | X |
| Control group (flush as needed) | | | | | | | | |
| Daily checklist for chest drain blockage assessment (states, patency, blockage) | | X | X | X | X | X | | |
| 20 mL of saline rescue flush if the chest drain is clogged | | X | X | X | X | X | | |
| Daily checklist for chest drain removal assessment | | X | X | X | X | X | | |
| Follow up | | | | | | | | X |

*If a chest CT scan is available, there is no need for Chest X-ray

**Starting immediately upon randomization, or if recently flushed, 6 ± 2 hours after the chest drain was flushed

**Fig 2. Study events and activities.** *If a chest CT scan is available, there is no need for chest X-ray. ** Starting immediately upon randomization, or if recently flushed, 6 ±2 hours after the chest drain was flushed.

Demographic data – age, gender, race, smoking status

1. Indication for chest drain placement.

2. Date and time of drain placement.

3. Amount of daily fluid drained while in place.

4. Imaging data (chest X-ray and ultrasound images).

5. Length of hospitalization.

6. Any instances of deviations from an assigned group (e.g., if a drain was thought clogged and underwent saline flushing outside of the protocol).

7. Replacement of drain secondary to dislodgement or inability to drain.

8. Presence or absence of continuous wall suction application (for >75% of the previous 24 hours).

9. Date and time of drain removal.

10. Data related to daily chest drain blockage assessment and chest drain removal objective data.

11. Categorization of daily drain status, meeting either "successful," "ongoing" or "failed" criteria.

12. Imaging data on day of chest drain removal (ultrasound, chest-X-ray, or CT scan) characterizing effusion status (Table 2).

Follow up-Data

1. Other treatment strategies: including presence of additional chest drain in the same pleural space, surgical consult (and if applicable surgery date and interventions).

2. Post hospitalization data: Including date of discharge, status of empyema 4–6 weeks and 3 months after removal, and date of infection resolution on chest imaging.

3. Complications: Including procedural and treatment related complications, or study-related complications.

4. Patient status at follow-up: To further characterize patient outcomes following discharge after successful chest drain removal, additional data will be collected regarding patient status at their follow-up visit and characterized based on five scenarios:

A) Follow-up: resolved infection: Upon clinician follow-up there is no further evidence of residual infection clinically and on imaging and no further need for evaluation or treatment for infection resolution.

B) Follow-up: surveillance: Upon clinician outpatient follow-up there is clinical concern for potential continued or residual infection despite initial chest drain drainage, requiring further clinical evaluation and potentially further procedures for resolution.

**Table 2. Imaging data collected on day of chest drain removal.**

| Day of Chest Drain Removal: Effusion Classification |
| --- |
| **Chest X-Ray** |
| 0 = no pleural fluid present |
| 1 = blunting of the costophrenic angle |
| 2 = fluid occupying up to 25% of the hemithorax |
| 3 = fluid occupying between 26–50% of the hemithorax |
| 4 = fluid occupying between 51–75% of the hemithorax |
| 5 = fluid occupying between 76–100% of the hemithorax |
| **Ultrasound** |
| Number of rib spaces occupied by effusion in the: |
| -Anterior view (#) |
| -Axillary view (#) |
| -Posterior view (#) |
| **CT Scan** |
| Effusion characteristics: |
| -Presence or absence of loculations |
| Effusion size in non-loculated effusion [12]: |
| - Scant volume/small |
| - ≤ 1/4 of hemithorax |
| - ¼ to ≤½ of hemithorax |
| - ½ to ≤ 3/4 of hemithorax |
| - ¾ to almost entire hemithorax |

C) Follow-up: Ongoing infection: Upon clinical outpatient follow-up there is clear imaging or clinical signs of ongoing pleural infection despite initial chest drain drainage, actively requiring further management.

D) Patient lost to follow-up: Following hospital discharge, the patient has been unable to be contacted or is not presenting to out-patient appointments.

E) Patient is deceased: Following hospital discharge, the patient is deceased. If available, cause of death will be documented.

## Data and safety monitoring board

The Data and Safety Monitoring Board (DSMB) will not be required for this trial. We do not expect additional safety concerns from this protocol over those incurred during conventional placement and maintenance of chest drains. These risks, inherent to the chest drain procedure itself, are discussed as part of the clinical informed consent and include bleeding, infection, pain, and pneumothorax. No interventions outside the standard of care are performed in this study.

Therefore, data, including adverse events related to standard of care will be collected. Adverse events related to research procedures (the consent process, HIPAA compliance, etc.) will be collected. This study is expected to have minimal to no adverse advents. This study is not anticipated to have any serious adverse events.

## Statistical analysis plan

Continuous variables will be summarized using the mean (SD) or median (range), as appropriate. Frequencies and percentages will be used to summarize categorical variables. The Wilcoxon test will be used to compare continuous variables between two different groups when covariate adjustment is not needed. The Pearson chi-square test will be applied to assess the association between two categorical variables such as complications by treatment. Cause-specific hazard models will be explored as sensitivity analyses. The proportional hazards assumption will be evaluated using Schoenfeld residuals and time-varying coefficients where appropriate.

Competing events for the primary endpoint include death with a chest drain in situ, chest drain dislodgement, malposition, or kinking. These events preclude observation of chest drain removal in the setting of treatment success. Censoring will occur only for withdrawal of consent prior to experiencing either chest drain removal and on rare occasions, lost to follow up in a patient who is discharged home or leaves against medical advice, with the chest tube in situ.

**Primary endpoint.** The primary endpoint analysis will use a competing-risks time-to-event framework to compare time from randomization to chest drain removal between treatment arms, accounting for clinically relevant competing event. The Wilcoxon rank sum test will be used if no competing events are documented within the final dataset.

**Secondary endpoints.** In addition to the primary endpoint of interest, we are interested in the comparison of the time to failure of therapy/drainage between two arms. For the time to failure of therapy/drainage analysis, Cox regression with adjustment for covariates of interest will be used. Adjusted models will include prespecified baseline covariates selected for clinical relevance and known prognostic importance, including age, baseline effusion complexity, and intrapleural enzyme therapy use. The number of selected covariates will be regulated to preserve model stability. The cumulative failure of therapy/drainage curve will be calculated from the Kaplan-Meier method and compared using the log-rank test. For the exploratory study, we will apply Markov ordinal longitudinal proportional odds state transition model to analyze our interventions into impact on the process of chest drainage to successful outcome or failure. Specifically, we will compute the estimated time in states 5, 6, 7 and 8 in each arm to further assess the role of chest drain flushing on number of days in a state with a clogged tube and with relation to their final outcome in an absorbing state. The times are sums of state occupancy probabilities [13]. This model will provide further characterization of intervention and control arms with regards to state of chest drain and state of patient among all populations, including those with chest drain failure, to characterize differences among these groups. Missingness on the primary or secondary outcomes is unlikely due to the proximity of

their measurement with the intervention and its integration into daily clinical documentation. Administrative censoring is expected to be minimal and will be monitored. No imputation is planned for time-to-event outcomes. There is no plan for interim analyses and no boundary for early stopping.

### Dissemination plans

After study completion and data analysis, any manuscript or releases resulting from the collaborative research must be approved by the investigators and will be circulated to applicable participating investigators prior to submission for publication or presentation. All data will be made available to all authors as required. Publication of results will be determined by the investigators. All authors are expected to disclose financial relationships or affiliations that could be considered conflicts of interest per journal or medical society requirements.

The results will be made available in ClinicalTrials.gov. Authorship will be consistent with the International Committee of Medical Journal Editors (ICMJE) guidelines.

### Discussion

Chest Drain REgular FLushing in ComplIcated Parapneumonic EFfusions and Empyemas (RELIEF) is the first randomized controlled trial comparing chest drain flushing protocols in small-bore chest drains (8Fr to 20Fr), the first line treatment for pleural infection management. While widely adopted as standard of care treatment for empyema and complicated parapneumonic effusion, once a chest drain is placed, centers vary in flushing practices, some at regular frequencies and some on an as need basis. Studying a standardized chest drain flushing practice can establish the role of flushing on pleural infection treatment outcomes, as well as promote the standardization of chest drain management practices in future studies of pleural space infection.

This study was designed to integrate each comparator arm into the regular clinical practice of chest drain management with ease. There is minimal additional disruption in patient care in either comparator arm, as both arms (regular chest drain flushing, as-needed flushing) currently represent the varied standard of care practices across centers.

We selected time to chest drain removal as our primary outcome due to its dual reflection of being a patient-centered and clinically relevant endpoint. Chest drains can be painful, mobility-limiting devices that in most cases, require frequent attention and a prolonged inpatient hospital stay. Earlier removal, in turn, may reduce hospital stays, costs, as well as patient morbidity. To clinicians, chest drain removal either indicates the successful source control of a clinically resolving pleural infection, or the need for an alternate treatment modality when the failure of an index chest drain is apparent. We recognize these varying outcomes and specifically utilized this to inform the most appropriate mode of analysis as a result.

In addition to the direct analysis of our primary outcome, we will apply multistate transition modeling to our time-to-event analysis for our study. Multistate transition modeling is an approach to longitudinal data analysis applied to clinical processes which reflect disease state progression over time, with the ability to quantify when treatment is most effective, while accommodating absorbing events [13]. We feel this is a novel and powerful tool to analyze dynamic disease processes and model the progression of a disease, influenced by various factors, from its initial stage, to progress in treatment, relapse or treatment completion, considering transitions between different states. To our knowledge, there are no studies that have applied multistate transition modeling to pleural disease research questions.

There are several potential limitations to this study. First, this is a non-blinded study, and those within the intervention arm of regular chest drain flushing receive more attention to their chest drain by default, with increased likelihood of noticing abnormalities, malfunction, or clogging. To address this, we collect daily data regarding the providers date and time of assessment of treatment "success" indicated by "readiness for chest drain removal," as well as the actual date and time of chest drain removal. This data will be used for sensitivity analysis to assess the role of non-blinding on time to remove the chest drain. Additionally, we are solely assessing the role of flushing in chest drains placed for pleural infection, which limits the generalizability of our findings to those placed, or remaining in place for alternate indications, such as

pneumothorax. Finally, while we highly encourage and document the usage of −20 cm $H_2O$ of wall suction in each comparator arm, we do not exclude patients that cannot tolerate suction, and while pragmatic in nature, recognize this as a potential limitation.

In summary, RELIEF is a multi-center, open label randomized controlled trial comparing a regular saline flushing protocol to as needed flushing in patients with chest drains placed for pleural space infection.

## Supporting information

**S1 File. Protocol Revised Final.**
(DOCX)

**S1 Checklist. Relief spirit 2025 checklist.**
(DOCX)

## Author contributions

**Conceptualization:** Jennifer D. Duke, Robert J. Lentz, Fabien Maldonado, Najib M. Rahman, Samira Shojaee.

**Data curation:** Jennifer D. Duke.

**Funding acquisition:** Samira Shojaee.

**Investigation:** Taryn K. Boyle, Jennifer D. Duke, Rafael Paez, Greta Bridwell, Ankush P. Ratwani, Kaele M. Leonard, Robert J. Lentz, Fabien Maldonado, Najib M. Rahman, Samira Shojaee.

**Methodology:** Taryn K. Boyle, Jennifer D. Duke, Heidi Chen, Frank E. Harrell Jr, Robert J. Lentz, Fabien Maldonado, Najib M. Rahman, Samira Shojaee.

**Project administration:** Taryn K. Boyle, Gulmira Yermakhanova, Samira Shojaee.

**Supervision:** Jennifer D. Duke, Samira Shojaee.

**Writing – original draft:** Taryn K. Boyle, Jennifer D. Duke, Gulmira Yermakhanova, Rafael Paez, Greta Bridwell, Ankush P. Ratwani, Kaele M. Leonard, Heidi Chen, Frank E. Harrell Jr, Robert J. Lentz, Fabien Maldonado, Najib M. Rahman, Samira Shojaee.

**Writing – review & editing:** Taryn K. Boyle, Jennifer D. Duke, Gulmira Yermakhanova, Rafael Paez, Greta Bridwell, Ankush P. Ratwani, Kaele M. Leonard, Heidi Chen, Frank E. Harrell Jr, Robert J. Lentz, Fabien Maldonado, Najib M. Rahman, Samira Shojaee.

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
