## [Decision Letter · Decision Letter 0]

10 Nov 2025

Dear Dr. Shojaee,

Thank you for submitting your manuscript to PLOS ONE. After careful consideration, we feel that it has merit but does not fully meet PLOS ONE’s publication criteria as it currently stands. Therefore, we invite you to submit a revised version of the manuscript that addresses the points raised during the review process.

We look forward to receiving your revised manuscript.

Kind regards,

Erik Su

Academic Editor

PLOS ONE

Journal Requirements:

2. We note that the original protocol that you have uploaded as a Supporting Information file contains an institutional logo. As this logo is likely copyrighted, we ask that you please remove it from this file and upload an updated version upon resubmission.

“This study is funded by Cook Medical, Inc.“

“Samira Shojaee: Research funding from Cook Medical. Fabien Maldonado: consulting for Medtronic, Intuitive, and J&J, Principal Investigator for the ACES study with research funding from Pleura Dynamics. Research funding from Medtronic. Robert Lentz: consulting for Intuitive.  Jennifer Duke: consulting for Intuitive, research funding from Cook Medical.”

Reviewers' comments:

Reviewer's Responses to Questions

**Comments to the Author**

1. Does the manuscript provide a valid rationale for the proposed study, with clearly identified and justified research questions?

Reviewer #1: Yes

Reviewer #2: Yes

2. Is the protocol technically sound and planned in a manner that will lead to a meaningful outcome and allow testing the stated hypotheses?

Reviewer #1: Yes

Reviewer #2: Yes

3. Is the methodology feasible and described in sufficient detail to allow the work to be replicable?

Reviewer #1: Yes

Reviewer #2: Yes

4. Have the authors described where all data underlying the findings will be made available when the study is complete?

Reviewer #1: Yes

Reviewer #2: Yes

5. Is the manuscript presented in an intelligible fashion and written in standard English?

Reviewer #1: Yes

Reviewer #2: Yes

You may also provide optional suggestions and comments to authors that they might find helpful in planning their study.

Reviewer #1: Limiting the Wilcoxon comparison to patients with “successful removal” excludes those with competing events and biases toward easier cases.

Proposing Wilcoxon first and competing-risks regression only “if needed” is inappropriate—competing events are expected.

Perhaps better to specify the competing-risks model as the primary method; do not make it conditional.

The plan does not clearly distinguish censoring from competing events. Define censoring clearly.

Specify effect measures (e.g., hazard ratio with 95% CI).

No plan for checking model assumptions, such as proportional hazard.

Stating that missing is “not expected” is not sufficient, especially since withdrawal or intrapleural enzyme-therapy failure can prevent reaching the primary endpoint.

“Adjustment of covariates of interest” is vague.

Reviewer #2: The authors present a very organized, well-planned, and thorough study protocol to determine whether routine saline flushes vs. no saline flushes will improve chest tube removal outcomes.

I think putting an explicit hard cut off of "small size chest tube" should be included in your inclusion/exclusion criteria, whether thats 12F, 14F or whatever size the authors deem to be appropriate.

I think including the indication for chest tube placement would be a great data and information to gather (empyema s/p CABG or VATS or etc.).

Though difficult, if the authors could included a standardized method of where the chest tube is placed and the method in which it is placed, this would greatly improve the impact of the study but I know that can be very difficult to standardize at any institution.

**Do you want your identity to be public for this peer review?** For information about this choice, including consent withdrawal, please see our Privacy Policy

Reviewer #1: No

Reviewer #2: No

---

## [Author Response · Author response to Decision Letter 1]

16 Jan 2026

December 15, 2025

Dear Editors and Reviewers ,

Thank you for reviewing our submission to PLOS ONE. Please see the revised manuscript attached as well as this itemized list of adjustments to the documents to better align with the journal style and the reviewer’s comments. We appreciate the reviewer’s thoughtful feedback, which has strengthened the methodological rigor and transparency of the protocol. All changes have been incorporated into the revised manuscript.

We note that the original protocol that you have uploaded as a Supporting Information file contains an institutional logo. As this logo is likely copyrighted, we ask that you please remove it from this file and upload an updated version upon resubmission.

The IRB approved protocol was updated with the logo removed.

Thank you for stating the following financial disclosure:

“This study is funded by Cook Medical, Inc.“

The language in the cover letter and manuscript have been updated per request to provide better clarity.

Thank you for stating the following in the Competing Interests section:

“Samira Shojaee: Research funding from Cook Medical. Fabien Maldonado: consulting for Medtronic, Intuitive, and J&J, Principal Investigator for the ACES study with research funding from Pleura Dynamics. Research funding from Medtronic. Robert Lentz: consulting for Intuitive. Jennifer Duke: consulting for Intuitive, research funding from Cook Medical.”

The language in the cover letter and manuscript have been updated per request to provide better clarity.

We have updated the competing interest statement in our cover letter. The competing interests reported in the manuscript do not alter the adherence to PLOS ONE policies.

PLOS requires an ORCID iD for the corresponding author in Editorial Manager on papers submitted after December 6th, 2016. Please ensure that you have an ORCID iD and that it is validated in Editorial Manager. To do this, go to ‘Update my Information’ (in the upper left-hand corner of the main menu), and click on the Fetch/Validate link next to the ORCID field. This will take you to the ORCID site and allow you to create a new iD or authenticate a pre-existing iD in Editorial Manager.

We have included the ORCID iD for the corresponding author in manuscript (ORCID ID: 0000-0001-9228-209X).

Please include captions for your Supporting Information files at the end of your manuscript, and update any in-text citations to match accordingly. Please see our Supporting Information guidelines for more information: http://journals.plos.org/plosone/s/supporting-information.

Thank you, this was updated.

We appreciate the recommendation.

We have reviewed our reference list and ensure that they are complete and correct.

Reviewers' comments:

Comments to the Author

Reviewer #1: Limiting the Wilcoxon comparison to patients with “successful removal” excludes those with competing events and biases toward easier cases.

Proposing Wilcoxon first and competing-risks regression only “if needed” is inappropriate—competing events are expected.

We agree and have revised the manuscript to specify competing-risks regression as the primary analytic approach for the primary endpoint. Given the clinical context of pleural infection, competing events such as death, chest drain dislodgement, malposition or kinking, are possible and were prespecified. Conditional language suggesting that competing-risks analysis would be applied only “if needed” has been removed.

Perhaps better to specify the competing-risks model as the primary method; do not make it conditional.

We have updated the language to ensure specific analysis. We agree that because competing events are expected in pleural infection management, competing-risks regression will be the primary analytic approach for the primary endpoint.

The plan does not clearly distinguish censoring from competing events. Define censoring clearly.

We thank the reviewer for highlighting this need for clarification. The revised manuscript now explicitly defines censoring. Competing events include death with a chest drain in situ, chest drain dislodgement or malfunction, malposition or kinking, . Censoring will occur in case of withdrawal of consent prior to experiencing chest drain removal, and rare occurrences such as loss to follow up such as leaving against medical advice with chest tube in situ.

Specify effect measures (e.g., hazard ratio with 95% CI).

We have revised the Statistical Analysis Plan to explicitly state the effect measure for the primary analysis. Treatment effects will be summarized using subdistribution hazard ratios (sHRs) with 95% confidence intervals derived from Fine–Gray competing-risks regression models.

No plan for checking model assumptions, such as proportional hazard.

We agree and have added language describing assessment of model assumptions. Specifically, the proportional hazards assumption will be evaluated using Schoenfeld residuals and time-varying coefficients where appropriate.

Stating that missing is “not expected” is not sufficient, especially since withdrawal or intrapleural enzyme-therapy failure can prevent reaching the primary endpoint.

We agree and have revised the manuscript to more accurately reflect anticipated data structure. Incomplete observation of the primary endpoint due to circumstances such as withdrawal, or death, will be handled analytically as censored data and competing events respectively, rather than treated as missing data. Administrative censoring is expected to be minimal. No imputation is planned for time-to-event outcomes.

“Adjustment of covariates of interest” is vague.

We have clarified the covariate adjustment strategy. Adjusted models will include prespecified baseline covariates selected for clinical relevance and known prognostic importance, including age, baseline effusion complexity, and intrapleural enzyme therapy use. The number of covariates will be limited to preserve model stability.

Reviewer #2: The authors present a very organized, well-planned, and thorough study protocol to determine whether routine saline flushes vs. no saline flushes will improve chest tube removal outcomes.

We appreciate your review of the paper.

I think putting an explicit hard cut off of "small size chest tube" should be included in your inclusion/exclusion criteria, whether thats 12F, 14F or whatever size the authors deem to be appropriate.

Thank you for this comment. The chest tube size has been included(8-20 Fr) to provide clarity. We have addressed this in two portions of the manuscript.

I think including the indication for chest tube placement would be a great data and information to gather (empyema s/p CABG or VATS or etc.).

Thank you for this comment. Per protocol, patients can be enrolled if they have a chest tubes placed for the indication of pleural space infection management. However, we have not obtained other specific causes of this etiology, such as for example, empyema due to transpleural spread of pneumonia, vs sepsis, and otherwise. We agree that this information would be of interest but felt that the details of the cause of empyema may not significantly impact the primary outcome of time to chest tube removal and thus have not collected this data. A comprehensive list of patients comorbid conditions are however collected.

Though difficult, if the authors could included a standardized method of where the chest tube is placed and the method in which it is placed, this would greatly improve the impact of the study but I know that can be very difficult to standardize at any institution.

Thank you for this comment. As the reviewer points out, we refrained from protocolizing chest tube placement methods and employed a more pragmatic approach to represent real-life practice across various centers. Chest tube method of placement and drainage is certainly of interest as it can impact the need for additional drainage catheters in the same pleural space. We have collected this data in the follow-up information. We hope this sheds light on incomplete drainage of a single catheter but also allows for generalizability of the results.

Please reach out with any additional questions or concerns.

Sincerely,

Corresponding author (on behalf of all authors)

---

## [Decision Letter · Decision Letter 1]

5 Feb 2026

Chest Drain REgular FLushing in ComplIcated Parapneumonic EFfusions and Empyemas: study protocol for the RELIEF randomized controlled trial

PONE-D-25-43311R1

Dear Dr. Shojaee,

We’re pleased to inform you that your manuscript has been judged scientifically suitable for publication and will be formally accepted for publication once it meets all outstanding technical requirements.

Kind regards,

Erik Su

Academic Editor

PLOS One

Additional Editor Comments (optional):

Reviewers' comments:

Reviewer's Responses to Questions

**Comments to the Author**

1. Does the manuscript provide a valid rationale for the proposed study, with clearly identified and justified research questions?

Reviewer #1: Yes

2. Is the protocol technically sound and planned in a manner that will lead to a meaningful outcome and allow testing the stated hypotheses?

Reviewer #1: Yes

3. Is the methodology feasible and described in sufficient detail to allow the work to be replicable?

Reviewer #1: Yes

4. Have the authors described where all data underlying the findings will be made available when the study is complete?

Reviewer #1: Yes

5. Is the manuscript presented in an intelligible fashion and written in standard English?

Reviewer #1: Yes

You may also provide optional suggestions and comments to authors that they might find helpful in planning their study.

Reviewer #1: All concerns are addressed.

**Do you want your identity to be public for this peer review?** For information about this choice, including consent withdrawal, please see our Privacy Policy

Reviewer #1: No

---

## [Editor Report · Acceptance letter]

PONE-D-25-43311R1

PLOS One

Dear Dr. Shojaee,

I'm pleased to inform you that your manuscript has been deemed suitable for publication in PLOS One. Congratulations! Your manuscript is now being handed over to our production team.

Kind regards,

on behalf of

Dr. Erik Su

Academic Editor

PLOS One